# Targeting a future generation free from female genital mutilation: A mixed-methods quasi-experimental study of an awareness intervention in central Tanzania

Leah Barthalome Kimario[1,2]*, Manji Nyaganya Isack[3], Beatrice Zakaria Temba[4], Agnes Cyril Msoka[2], Blandina Theophil Mmbaga[2]

1 Department of Nursing, Kilimanjaro College of Health and Allied Sciences, Moshi, Tanzania,
2 Kilimanjaro Clinical Research Institute, Moshi, Tanzania, 3 Department of Nursing, Kilimanjaro Christian Medical Centre, Moshi, Tanzania, 4 Department of Public Health, Kilimanjaro Christian Medical University College, Moshi, Tanzania

* kimarioleah@yahoo.com

## Abstract

Female genital mutilation (FGM) persists in Tanzania, with the Dodoma Region having the country's second-highest prevalence. This study evaluated a community-based educational intervention aimed at increasing FGM awareness and encouraging its abandonment among young adults (15–19 years) in Chamwino District, Dodoma, an important yet often overlooked group for fostering intergenerational change. We conducted a mixed-methods study with a primary quasi-experimental component: a single-arm, baseline-to-endline survey with a multi-stage cluster sample of 452 young adults who completed follow-up from schools and hospitals. Secondary components included a clinical audit of 3,770 delivering mothers to determine FGM prevalence and in-depth interviews with ten young adult mothers who had undergone FGM. The primary outcome was the change in the mean awareness score, analyzed using a paired t-test, with linear regression estimating associations with participants' demographics. Changes in binary secondary outcome measures were analyzed using McNemar's test, and the association between them was assessed with the chi-square test. The mean awareness score increased by 15.04 percentage points from 44.08% (SD = 15.85) at baseline to 59.12% (SD = 16.85), p < .001. At endline, 96.5% believed FGM is harmful, 50.9% reported its continuation, 97.6% knew it violates the human rights of girls and women, and 95.8% expressed a desire for abandonment. About 7.3% of women self-reported positive FGM status at baseline, while 16.6% were observed in 9-months. Qualitative analysis yielded three themes: 1) The hidden system, 2) Blood and lies, and 3) Intergenerational revolt. This theory-informed, community-engaged intervention significantly improved awareness and attitudes among more than 1,700 community members. However, qualitative findings revealed a persistent "hidden system" and a "critical power gap" among young

**Data availability statement:** All aggregated data underlying the findings reported in this paper are presented within the main text, tables, figures, and Supporting information files. Raw individual-level data (qualitative transcripts, individual survey responses, and facility-level audit data) contain potentially identifiable information about participants and institutions (schools, health facilities). Due to the sensitive nature of FGM and protecting the study population, public deposition of these raw data could compromise participant confidentiality and expose them to potential social or legal harm, and breach compliance with the protocol approved by the ethics review committee. Therefore, these raw datasets are not publicly available. The corresponding author holds the data. Queries regarding data access and reasonable requests for de-identified data should be directed to the College Research Ethics Review Committee (CRERC) Secretary at Kilimanjaro Christian Medical University College (email: info@kcmcu.ac.tz).

**Funding:** This project was funded by the GAIA Initiative Grant (Project 21-S22F to BTM). The funder had no role in the study design, data collection, analysis, reporting, decision to publish, or manuscript preparation.

**Competing interests:** The authors have declared that no competing interests exist.

**Abbreviations:** CI, Confidence Interval; FGM, Female Genital Mutilation; KI, Key Informant; SD, Standard Deviation; TDHS-MIS, Tanzania Demographic and Health Survey and Malaria Indicator Survey; WHO, World Health Organization.

adults, indicating awareness alone is insufficient. Sustainable abandonment requires integrating awareness campaigns with strategies that address structural power dynamics.

## Introduction

FGM affects over 230 million women and girls globally, mainly in Africa [1], with 8% of reproductive women in Tanzania affected [2]. Complications are severe and range from short-term to long-term, including urogynecologic, psychological, and obstetric [3], requiring ongoing efforts undertaken by this study to protect the next generation of women. Dodoma bears the country's second-highest FGM prevalence (47%). Practices are shifting to early childhood to evade laws [2,4], enduring due to deeply rooted socio-cultural norms and gender inequalities, despite legislation and awareness efforts [5].

Global guidance from the World Health Organization (WHO) emphasizes the importance of educational interventions targeting women, girls, men, and boys in FGM-affected communities as a core prevention strategy [6]. Systematic reviews of intervention effectiveness confirm that community-level approaches can shift attitudes, though translating these into behaviour change remains challenging [4]. A critical gap is the limited focus on children and young adults, who are influential, less bound by tradition, and more receptive to change [7,8]. Empowering them supports Sustainable Development Goal target 5.3 to eliminate harmful practices [9].

This gap is especially evident in high-prevalence, rural districts like Chamwino, a Gogo tribe heartland with high FGM rates [10,11]. Despite this, little is known about young adults' awareness, attitudes, and experiences regarding FGM in Chamwino, and no age-focused programs have been documented. This study aimed to evaluate a theory-informed, community-based educational intervention designed to increase awareness of FGM risks and promote abandonment among young adults. We hypothesized that participants in the multi-component educational intervention would demonstrate a significant increase in awareness of FGM health risks, proven by the mean awareness score (baseline-endline). Using mixed methods, we also explored the FGM prevalence and experiences to contextualize the intervention impact and identify persistent socio-structural barriers to abandonment.

## Materials and methods

### Ethics statement

This study was approved by the College Research Ethics Review Committee of Kilimanjaro Christian Medical University College (Cert. No. 2625). Permissions were obtained from local authorities and institutions. A tiered consent model was used: parental permission and student assent were obtained for school-based activities involving minors under 18; adults provided written consent. For the FGM prevalence audit, trained midwives obtained documented oral consent from mothers to ensure confidentiality and practicality in the high-volume clinical setting. For young mothers

(hospital survey/interviews), written consent was obtained; a waiver was granted for mature minors (ages 15–17) to protect them from forced disclosure and to account for the impracticality of contacting parents during short postpartum stays. Household visits required written permission from the head and verbal consent from members. Attendance at public gatherings and hospital training constituted consent, with voluntary participation. Psychological support was available at the institutional level in all study activities. Throughout the study, no adverse events were reported. Specifically, no participants experienced psychological distress requiring referral, social backlash (e.g., family conflict or community stigmatization), or breaches of confidentiality. Champions and researchers were trained to identify and report any such incidents, and none were documented. Confidentiality was maintained through the use of anonymized codes, secure storage, and deletion of identifying links after analysis.

## Study design and registration

This was a community-based, mixed-methods study conducted over one year from April 2023 to March 2024. The primary component was a single-arm, baseline-endline quasi-experimental design (non-randomized) to evaluate a multi-component educational intervention. The other components were: a prospective clinical audit of FGM prevalence to provide contextual depth; qualitative interviews to explain the quantitative findings; and a process evaluation of implementation feasibility and reach. The study protocol was finalized before participant enrollment and retrospectively registered with the Open Science Framework (OSF) [https://doi.org/10.17605/OSF.IO/XCQRT]. Reporting of quantitative findings followed the Transparent Reporting of Evaluations with Nonrandomized Designs (TREND Statement Checklist) S1 Checklist [12] and qualitative findings followed the Consolidated Criteria for Reporting Qualitative Research (COREQ) S2 Checklist [13]. The study proceeded in four phases: (1) Preparation and baseline (April-June 2023): obtaining approvals, validating research/intervention instruments, and starting longitudinal data collection with a baseline survey and prevalence data; (2) Intervention initiation (June-August 2023): training community champions, forming an evaluation team, and starting qualitative interviews; (3) Implementation and monitoring: deploying intervention activities across schools, hospitals, and communities while continuing prevalence and qualitative data collection; and (4) Endline and evaluation (March 2024): endline survey, final analysis, and community feedback.

## Theoretical framework

The intervention integrated behavioral models to address the multifaceted drivers of FGM. The Health Belief Model shaped perceptions of health risks and perceived benefits of abandonment [14]. Social Cognitive Theory employed peer champions and testimonials to build self-efficacy and model new norms [15]. The Theory of Planned Behaviour targeted attitudes, subjective norms (social pressure), and perceived control [16]. The Socio-Ecological Model structured the intervention across levels: individual (awareness), interpersonal (family dialogues), community (mobilization), and societal (laws and gender norms) [17].

## Study setting and participants

This project was conducted in Chamwino District, a high-prevalence area of FGM in the Dodoma Region of Central Tanzania. Besides the Gogo, FGM in Chamwino is practiced across multiple tribes, including both indigenous and migrant groups such as Sandawe, Burunge, Rangi, Maasai, and many others. FGM remains discreet, with targeted interventions lacking in key population groups (e.g., children, young adults). In 2022, Chamwino's population accounted for approximately 16% of Dodoma's population (486,176 of 3,085,625), including 49,807 young adults [18]. Six of 36 district wards were purposively selected to ensure urban/rural diversity, covering known variations in FGM prevalence and practical considerations related to budget, timeline, and geographical accessibility, given the district's ward dispersions. We aimed to include one secondary school (with at least 71 students aged 15–19) and one public health facility (with at least seven

deliveries among young adults in the past three consecutive months) to facilitate rapid recruitment. Low delivery rates at Itiso health facilities and during the holidays in the baseline survey at Mlowa and Mpwayungu Wards resulted in respective institutions lacking sample representation. Missed samples were evenly distributed across similar strata (e.g., school-for-schools, hospital-for-hospital), and analysis was adjusted accordingly. The final sample (5 hospitals and 4 secondary schools) and their corresponding baseline-endline survey participants are detailed in Table 1.

## Participants' sampling, recruitment, and eligibility criteria

Participant recruitment spanned 10 months, from the baseline survey (May 26–June 9, 2023) to the endline (March 1–15, 2024). FGM prevalence data were collected consecutively for 9 months (June 1, 2023–February 29, 2024), and qualitative interviews were conducted over 6 months (July 30, 2023–January 16, 2024). Sampling methods varied by component.

Baseline-endline participants were young adults aged 15–19 enrolled in four schools or receiving maternity services at five hospitals. To minimize loss to follow-up, individuals planning to leave the district, those in their final year of schooling, and individuals unable to consent were excluded. For young mothers, baseline data were collected during the postnatal hospital stay. Given the practical difficulty of tracking participants after discharge, follow-up strategies included linkage to scheduled postnatal visits and optional, participant-defined contact methods to facilitate endline data collection.

Multi-stage cluster sampling with probability proportional to size was used, assuming a 10:1 school-to-hospital eligible population ratio. The feasibility-based target (468 calculated sample) was 426 from schools and 42 from hospitals, balanced by gender and ward (78; 71 from schools and seven from hospitals). To meet the school target amid an anticipated ~30% non-response rate due to a sensitive topic, consent/assent forms were distributed to 610 students in randomly selected classes across four schools; 510 (83.6%) returned them. After verification, 42 students were excluded (age outside range: n = 18; residency <1 year: n = 16; other: n = 8), 426 were enrolled, matching the school target precisely. In hospitals, census sampling was used. No refusals were recorded among the 42 young mothers who delivered at baseline and were approached. Total baseline enrollment was 468, meeting the target, with 452 (96.6%) retained for paired analysis.

Qualitative purposive sampling selected 10 FGM-positive young mothers [15–19] who recently delivered at study hospitals, knew of ongoing FGM, and were willing to share experiences. This sample size, within a recommended range (5–25) for phenomenological studies [19] allowed in-depth exploration until thematic saturation at the phenomenon level [20]. Four declined due to fear. Those unable to consent, in pain, or unable to endure post-birth interviews were excluded.

The FGM prevalence audit included all delivering mothers at five hospitals. Over 9 months, 3,842 mothers delivered; 72 (1.9%) were excluded (missing consent: n = 48; refusal: n = 24), leaving 3,770 (98.1%). General inclusion and exclusion

**Table 1. Study wards, participating institutions, and corresponding baseline-endline participants.**

| Ward | Type | Representatives | | Participants/Ward | | | | | |
|------|------|-----------------|--|-------------------|--|--|--|--|--|
| | | Health Facility | Secondary school | Baseline | | | Endline | | |
| | | | | M | F | Total (n, %) | M | F | Total (n, %) |
| Mpwayungu | Remote | Mpwayungu Health Centre | — | — | 8 | 8 | — | 8 | 8 |
| Haneti | Remote | Haneti Health Centre | Haneti | 59 | 57 | 116 | 57 | 53 | 110 |
| Chamwino | Semi-urban | Chamwino Health Centre | Chamwino | 59 | 57 | 116 | 59 | 56 | 115 |
| Itiso | Remote | — | Itiso | 58 | 48 | 106 | 54 | 46 | 100 |
| Mlowa | Semi-urban | Chamwino District Hospital | — | — | 8 | 8 | — | 8 | 8 |
| Dabalo | Remote | Dabalo Health Centre | Dabalo | 58 | 56 | 114 | 58 | 53 | 111 |
| Total (n, %) | | 5 | 4 | 234 (50) | 234 (50) | 468 (100) | 228 (50.4) | 224 (49.6) | 452 (100) |

criteria for the baseline-endline survey and interview: participants had to have lived in the district for at least 1 year to ensure sufficient exposure. Those unable to consent, cognitively impaired, or with less than a year of residence were excluded. The participant flow is detailed in Fig 1.

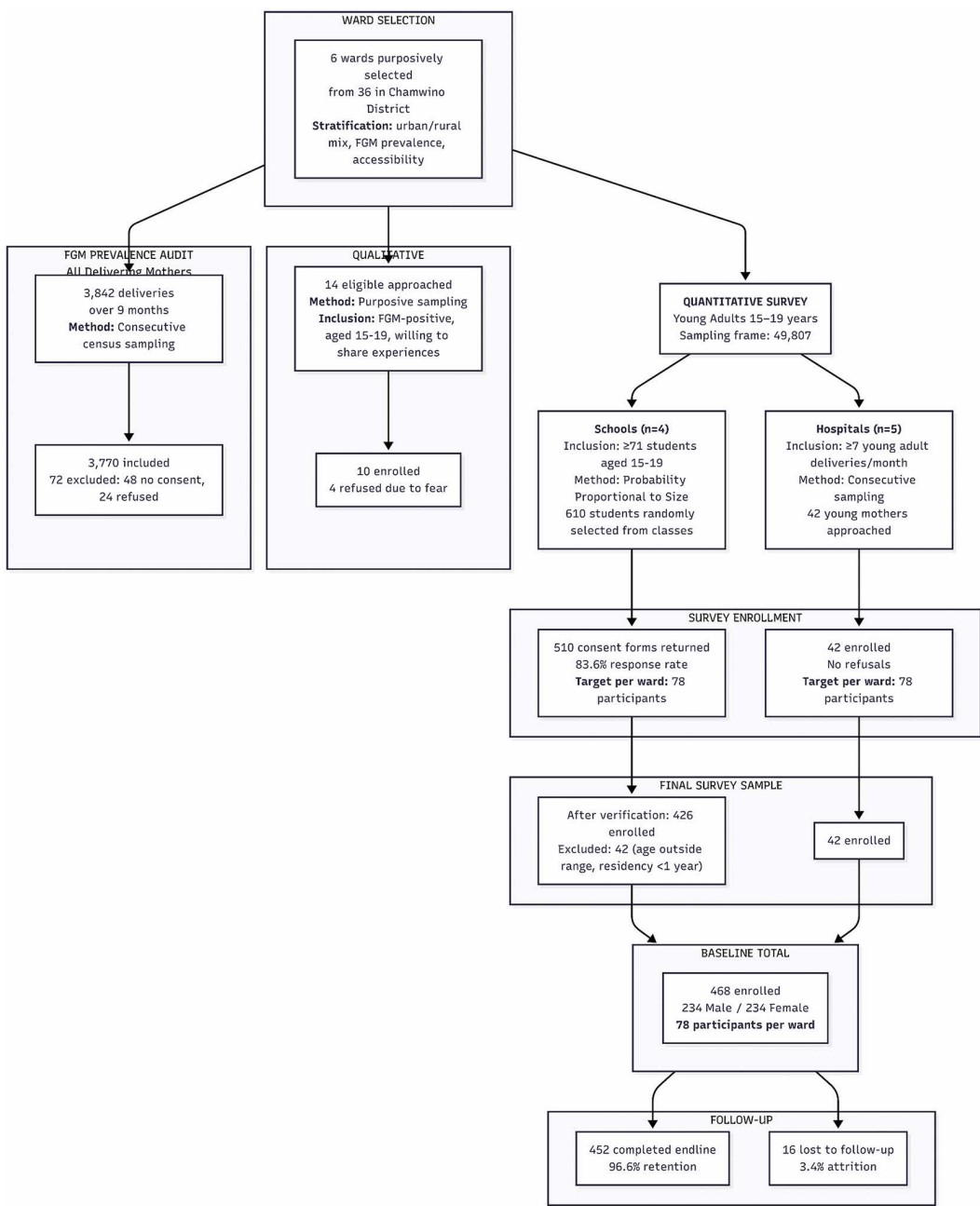

**Fig 1. Participant recruitment flow diagram for the mixed-methods study.**

## Interventions

The 6-month intervention (September 2023-February 2024) was a multi-component, community-based educational campaign, following the literature-based training manual S1 Text [2,21–24], informed by the four integrated behavioral theories [14–17]. These theories served as independent variables in the quasi-experimental design, allowing empirical assessment of their causal impact on FGM awareness among a previously unresearched, intervention-naïve population.

Fifty champions, 15 adults from (health workers, teachers, local/religious leaders, elders, traditional birth attendants, community health workers), and 35 students were trained. An 11-member evaluation team, including the two lead authors, representatives from the champion groups, and a police officer, had three key roles: (1) validated data collection instruments, training manual, and intervention activities for cultural and contextual fit, (2) monitored the intervention's impact and activities, and (3) set sustainable goals. Implementation of activities was ward-based, guided by baseline needs, and adapted to the district's dispersed geography. The core educational content from the validated manual was consistently implemented.

Educational sessions were delivered across multiple settings. In schools, 737 students participated in initial gender-stratified sessions. Student and teacher champions collaborated to establish health clubs and debate sessions. Health clubs, with 35 participants (totaling 105 attendances), met three times at one school, and four debate sessions were held in two schools without gender stratification to promote shared perspectives. Attendance at the open-access debate sessions was not formally recorded because they were voluntary forums for open dialogue, prioritizing discussion over participant counts.

In hospitals, 13 training sessions were delivered in labor wards and antenatal clinics to 246 participants. In line with WHO recommendations for strengthening health worker capacity in FGM prevention and care [6], nurse-midwives received additional training on classifying FGM typology and conducting educational sessions, using visual aids to improve clinical documentation and management of complications. Approximately 80% of nurse-midwives were trained to maximize data capture across hospitals.

Community mobilization involved household visits and large gatherings. Guided by feasibility estimates, a team of 15, including 10 champions (7 students, 3 teachers) and the two lead authors, conducted 46 household visits over a month in one ward and 2 large gatherings in 2 wards, thereby facilitating broader, open dialogue. A local government representative accompanied all activities. All household members were invited, but the census was only for those 12 or older who could comprehend the educational message [25]. No households refused consent.

## Outcomes and measurement

The primary outcome was the change in the mean awareness score, reflecting the overall shift in factual awareness about FGM health risks. Awareness was assessed using a 19-point composite score derived from 12 questions with multiple sub-items (1 point per correct response). The score included items on: FGM definition (1 point), harmful effects (7 points), human rights violation status (1 point), specific types of violations (6 points), criminal status (1 point), awareness of the International Day (1 point), and desire for abandonment (1 point). These items were selected based on established frameworks of FGM-related knowledge, including WHO guidance on health risks, human rights violations, and legal status [6]. The instrument demonstrated acceptable internal consistency (Cronbach's $\alpha = 0.70$). The full instrument and detailed scoring key are provided in the S1 Data.

Items excluded from the score, sources of information, perceptions of ongoing practice, reasons for continuation, and project ownership were analyzed separately as descriptive measures, reflecting the conceptual distinction between factual awareness and social norms [26]. Sources of information measure exposure pathways, not factual awareness; perception of ongoing practice captures community norms, not awareness; reasons for continuation were open-ended and not reliably scorable; and project ownership was a process measure, not an awareness outcome. Self-reported FGM status and prior program involvement were also excluded, as they reflect personal history rather than current awareness.

To contextualize the primary findings and identify specific areas of change that might inform targeted intervention components, additional secondary exploratory analyses were conducted on four aggregated measures: awareness that FGM is harmful, violates human rights of girls and women, desire for abandonment (as indicators of factual awareness), and acknowledgment of whether FGM is still occurring in the community (as an indicator of perceived community norms).

Using a prospective clinical audit, the study also measured FGM prevalence among delivering women to provide a community-level context. This audit allowed us to: (1) assess the gap between self-reported and clinically observed FGM prevalence, revealing the hidden nature of the practice; (2) identify high-burden areas for targeted intervention; and (3) establish a baseline for future prevalence studies, including aggregated tribal data. The audit generated granular district-level data, disaggregated by young adults (15–19 years) and tribe, details not specifically portrayed in the 2022 Tanzania Demographic and Health Survey and Malaria Indicator Survey (TDHS-MIS) regional estimate of 47% for Dodoma [2]. TDHS-MIS data were not used as a baseline comparator because they are aggregated at the regional level, whereas our study focused specifically on Chamwino District, a sub-region with distinct socio-demographic characteristics.

## Qualitative component

A concurrent qualitative phenomenological design was used to explore the lived experiences, perceptions, and contextual barriers among FGM-positive young mothers. The purpose was to understand the physical, psychological, and social impacts from the perspective of those who have undergone the practice, and explain the quantitative findings in this previously unexplored community.

## Data collection

The mixed-methods instruments, based on literature about FGM prevalence, awareness, norms, and abandonment strategies [2,5,8,21–23], were translated into Swahili, validated, pretested, and refined in a non-participating ward among eligible participants. The awareness instrument was tested with 47 secondary school students (10% of 468), the interview guide was tested through two hospital interviews, and the FGM audit checklist was tested with labor ward staff (10 cases) to ensure accurate categorization and workflow integration.

Data collection procedures were tailored to each component. For the survey, administration (self- or interviewer-based) was adapted to literacy levels. School-based surveys were administered in open areas, with participants spaced 1 meter apart, to reduce social desirability bias and ensure privacy; completeness was verified on-site.

For FGM prevalence, trained nurse-midwives used a standardized checklist to collect primary data from all delivering mothers during care prospectively. Data quality was ensured through cross-verification against birth registries, specialized training in the WHO FGM typology, and supervision.

For qualitative, the two lead authors (LBK, MNI), motivated by FGM abandonment, bearing clinical and public health experience, conducted separate five private, face-to-face, audio-recorded in-depth interviews each in Swahili with FGM-positive young mothers in hospital labor wards, to reduce bias. Interviews lasted 30–60 minutes, followed member-checking at the end of each interview (transcripts were not returned to participants due to environmental impracticality), and were supplemented with field notes. Interviews were once conducted without repeats or a third person present.

## Data analysis

Quantitative analysis used SPSS v25 to evaluate intervention effects, providing detailed data on participant demographics and outcome distributions to inform stakeholders and guide future strategies. Given the single-arm, pre-post design, an intention-to-treat analysis was not applicable. Instead, the primary analysis followed a complete-case approach (n = 452). Normality of the awareness score distribution was confirmed using the Shapiro-Wilk test (W = 0.995, p = 0.130).

Primary outcome. The change in mean awareness score was analyzed using a paired t-test, with Cohen's d as the effect size. To examine factors associated with follow-up awareness scores, linear regression was used, with participant demographics entered as covariates. Results are reported as unstandardized beta coefficients (B) with 95% confidence intervals. Ward of residence, the only demographic which showed statistical significance, Haneti, with the highest burden of observed FGM prevalence, was selected as the reference category. This allowed comparison of all other wards against the highest-burden context, highlighting where awareness gains were most pronounced relative to the most affected area.

Secondary exploratory measures. Changes in four aggregated binary measures were analyzed using McNemar's test, with proportion change (net difference between baseline and endline) reported as the effect size. Bivariate associations between these binary measures at endline were examined using chi-square tests, with phi ($\varphi$) reported as effect sizes for 2 × 2 tables.

Contextual measures. Descriptive statistics summarized audited FGM prevalence, with data disaggregated by types, age group, facility, and tribe.

All inferential statistics employed a 5% significance level and reported effect measures with 95% confidence intervals, per contemporary reporting standards [27].

Qualitative analysis. Audio recordings were transcribed verbatim, translated into English, and analyzed manually following Braun and Clarke's inductive thematic analysis framework [28]. To ensure analytical rigor and mitigate bias, two researchers (LBK, MNI) first independently familiarized and coded the same five transcripts. They then met in a series of discussions to compare, reconcile, and refine their codebooks into a consensus-based preliminary thematic framework. This framework was applied to the remaining transcripts, refined collaboratively, then reviewed and validated by the entire research team. Quantitative and qualitative findings were integrated via narrative synthesis in the discussion.

## Results

### Intervention reach, fidelity, and process evaluation

The campaign directly engaged over 1,700 individuals, including more than 900 students, 246 in hospital settings, 104 in households, and over 500 attendees at the two community gatherings. Process evaluation indicated strong implementation fidelity and high participant engagement, with trained champions leading activities. Fidelity was assessed through three criteria: (1) adherence to the validated training manual across all sessions; (2) delivery of core educational content by trained champions; and (3) consistent use of standardized materials as verified through field observations and champion reports.

Student champions served as change agents, delivering main speeches during household visits and community gatherings to promote ownership and message credibility, and were briefed in advance on the content and language. Health workers and traditional birth attendants shared testimonies in health clubs, with a police officer providing legal and reporting mechanisms. Anonymous feedback showed high project ownership with a mean proportion (84.7%) across 77.9% (household visits), 83.6% (community gatherings), and 92.5% (endline survey).

Health clubs were sustained post-study and evolved into broader health education platforms. Furthermore, a private WhatsApp group connected dispersed adult champions for remote supervision and peer support. Following the study, this developed into a sustained, community-led network for ongoing anti-FGM initiatives. Student champions, excluded due to school phone policies, were integrated through in-person meetings and final planning sessions.

### Participants' demographics

At the endline, among the 452 participants who completed follow-up, the median age was 17 years; 50.4% were male, and 93.1% had secondary education. Chamwino Ward contributed the most participants (25.4%). Most (72.8%) came from remote wards, and 91.6% were students. Of the 33 participant tribes, the majority were Gogo (57.5%). The median

duration of community residence was 16 years. These demographics are shown in Table 2, and a full list of participant tribes (baseline to endline) is provided in S1 Table.

## Attrition analysis

Analysis of baseline characteristics between participants retained for analysis (n = 452) and those lost to follow-up (n = 16) showed no statistically significant differences (all p > 0.05) in sex, age, ward of residence, type of ward, tribe, education level, self-reported FGM status among females, or, most importantly, mean awareness score. Attrition differed by participant group (p < 0.001), with more loss from schools (12/16) than hospitals, consistent with the greater logistical difficulty of tracking school-based participants over time. Overall, attrition was minimal (3.4%) and unlikely to have introduced systematic bias in the primary findings [29].

## Changes in FGM awareness score (baseline–endline)

The mean awareness score increased from 44.08% (SD = 15.85) at baseline to 59.12% (SD = 16.85) at follow-up, a mean increase of 15.04 percentage points (95% CI: 12.90, 17.18); t(451) = 13.81, p < .001, with a medium-to-large effect size

Table 2. Participants' demographics at baseline (N = 468) and endline (N = 452), Chamwino District, Tanzania.

| Variable | | Baseline (N = 468) | | | Endline (N = 452) | | |
|---|---|---|---|---|---|---|---|
| | | (n, %) | Subtotal (n, %) | Median (Range) | (n, %) | Subtotal (n, %) | Median (Range) |
| Age (years) | | | | | | | |
| 15 | | 89 (19.0) | | **17 (15–19)** | 84 (18.6) | | **17 (15–19)** |
| 16 | | 107 (22.9) | | | 105 (23.2) | | |
| 17 | | 112 (23.9) | | | 107 (23.7) | | |
| 18 | | 107 (22.9) | | | 104 (23.0) | | |
| 19 | | 53 (11.3) | | | 52 (11.5) | | |
| Years of community residence | | | | **16 (1-19)** | | | **16 (1-19)** |
| Sex, Ward of residence, and Ward type (refer to Table 1) | | | | | | | |
| Participant group | | | | | | | |
| Secondary school (students) | Itiso | 106 (22.6) | **426 (91.0)** | | 100 (22.1) | **414 (91.6)** | |
| | Chamwino | 107 (22.9) | | | 106 (23.5) | | |
| | Haneti | 107 (22.9) | | | 102 (22.6) | | |
| | Dabalo | 106 (22.6) | | | 106 (23.5) | | |
| Hospitals (young mothers) | Mpwayungu Health Centre | 9 (1.9) | **42 (9.0)** | | 8 (1.8) | **38 (8.4)** | |
| | Chamwino District Hospital | 8 (1.7) | | | 8 (1.8) | | |
| | Chamwino Health Centre | 9 (1.9) | | | 9 (2.0) | | |
| | Haneti Health Centre | 8 (1.7) | | | 8 (1.8) | | |
| | Dabalo Health Centre | 8 (1.7) | | | 5 (1.1) | | |
| Education level | | | | | | | |
| Secondary | | 433 (92.5) | | | 421(93.1) | | |
| Primary | | 35 (7.5) | | | 31 (6.9) | | |
| Tribe (n = 33) | | | | | | | |
| Gogo | | 267 (57.1) | | | 260 (57.5) | | |
| Others | | 201 (42.9) | | | 192 (42.5) | | |

(Cohen's d = 0.65). Baseline awareness did not predict endline scores (r = -0.001, p = 0.975), indicating that the intervention benefited participants across all levels, with an average improvement of 15.04 points and SD_diff of 23.16.

Linear regression of participants' demographics and follow-up awareness scores showed overall significance (F(11, 440) = 3.281, p < 0.001) and explained 7.6% of the variance in endline awareness scores (R² = 0.076). After adjusting for all covariates, ward of residence was significantly associated with awareness scores. Compared to Haneti, the ward with the highest observed FGM prevalence, participants from: Chamwino (B = 10.32, 95% CI: 5.58–15.06, p < 0.001), Dabalo (B = 9.45, 95% CI: 5.02–13.89, p < 0.001), Itiso (B = 6.29, 95% CI: 1.74–10.85, p = 0.007), and Mlowa (B = 18.11, 95% CI: 4.40–31.82, p = 0.010) scored significantly higher than participants from Haneti. Age, sex, education, years in the community, tribe, and participant group were not significantly associated with follow-up awareness scores (all p > 0.05).

**Changes in secondary exploratory binary measures (baseline–endline)**

Awareness that FGM is harmful increased from 91.2% to 96.7% (McNemar's χ² = 10.47, p = .001), a net increase of 25 individuals (40 gained awareness, 15 regressed). At endline, excessive bleeding was the commonly identified harmful effect of FGM by (91.5%) of the participants, followed by severe pain (81.7%) and infections (71.2%). About 92.7% believed FGM is conducted under no medical reasoning, and 93.8% knew the 'International Day for Zero Tolerance for FGM' and its core aims. School education was the primary source of FGM awareness (93.1%), followed by mass media (29.2%) and community discussions (16.2%). However, none of the participants reported prior involvement in an anti-FGM program at baseline, confirming a critical gap in targeted interventions, which the present study aimed to address.

Awareness that FGM violates human rights of girls and women increased from 85.0% to 97.6% (McNemar's χ² = 39.41, p = p < 0.001), a net increase of 54 individuals (64 gained awareness, 10 regressed). Among those recognizing it as a violation, the most cited associated abuses were child/early/forced marriage (61.7%) and intimate partner violence (58.4%). 96.7% knew FGM is criminalized by national and international laws.

Acknowledgment that FGM is still occurring in the community increased from 39.7% to 50.9% (McNemar's χ² = 9.65, p = .002), a net increase of 51 individuals (155 changed from 'No' to 'Yes', 104 regressed). Among those acknowledging, 60.9% could not specify reasons for its occurrence, although preserving norms and honoring ancestors were the most frequently cited reasons, by 17.4% (Table 3).

The desire to abandon FGM increased from 81.2% to 95.8% (McNemar's χ² = 40.63, p < .001), a net increase of 66 individuals (85 changed from 'No' to 'Yes', 19 regressed). At endline, bivariate associations between the four binary outcomes using chi-square tests showed that awareness that FGM is harmful was weakly associated with acknowledgment of its continuation (χ² [1] = 5.84, p = 0.016, φ = 0.114), and not significantly associated with desire for abandonment (p = 0.073) or with recognition of FGM as a human rights violation (p = 0.280). Table 4 presents the full aggregated data of the awareness score (baseline-endline). Additional aggregated data of the awareness score stratified by sex characteristics are provided as S2 Table.

**FGM prevalence and typology.** 3,770 delivering mothers aged 13–41 were observed for FGM status across 5 health facilities in Chamwino District over 9 months. The overall FGM prevalence was 16.6%, with a higher rate among young adults (31.0%). Most FGM cases were Type II (97.4%) and among the Gogo tribe (73.6%). The remote Haneti Ward had the highest FGM cases, at 51.5%, accounting for 43.3% of all district cases. Additionally, 7.3% self-reported a positive FGM status at baseline (Table 4 and 5).

**Qualitative findings: Lived experiences, systemic barriers, and participant-driven solutions**

The ten interviews recruited two participants from each of the five hospitals. Tribes: Seven were Gogo, with Maasai, Burunge, and Rangi (1 each). Ages: 15–16 (2), 17–18 (3 each), and 19 (2). Half were married or unmarried. Six had secondary education; four had primary education. The three main themes are: the hidden system, blood and lies, and intergenerational revolt, supported by six categories and 27 illustrative codes (Table 6).

**Table 3. Perceived reasons for the continuation of FGM among participants aware of its ongoing practice (n = 230) in the Chamwino District.**

| Reason for FGM occurrence in the Chamwino community | (n, %) |
|---|---|
| n = 230 | |
| Enhancing women's value during marriage | 5 (2.2) |
| Preventing stigma/earning respect | 4 (1.7) |
| Controlling women's sexuality | 13 (5.7) |
| Preserving norms and honoring ancestors | 40 (17.4) |
| Children's inability to question elders | 2 (0.9) |
| Rite to womanhood | 6 (2.6) |
| Treating vagina and urinary tract infections ("lawalawa") | 3 (1.3) |
| Do not know/ non-response | 140 (60.9) |
| Insufficient education on consequences | 14 (6.1) |
| Lack of community-tailored anti-FGM programs | 3 (1.3) |

**Theme 1: The hidden system.** FGM persists as a covert, adaptive system where public ceremonies have been abandoned in favor of secretive practices on infants and very young children to evade laws, making it difficult to determine the current rates. As participants described, "now babies are still cut discreetly very young, even before puerperium, is complete" (KI6); "knowing how much, currently, is hard…ceremonies are no longer held…" (KI8); as even "practitioners disguise themselves as traditional healers or birth attendants... their identities are kept classified among a few elders…" (KI10).

Specific gender dynamics reinforce this concealment. While elder women lead the practice, fathers sustain it through passive complicity, often dismissing FGM as a "woman's issue" (KI3) and upholding the belief it is necessary for marriage-ability, as they "dare not punish their wives or mothers" (KI9). This highlights a critical need for targeted interventions to build "fathers' self-efficacy and legitimize their decision-making power over elders' intentions" (KI8).

To maintain this hidden system, parents actively deceive children and avoid direct conversations, fearing that "if they find out early, they could expose it, cause trouble, or refuse" (KI1). Most (7/10) underwent FGM in early childhood and had no recall of the event. The remaining three, cut at ages 6–8, retained conscious memories. Notably, only three of the ten believed FGM made them "complete women." Deliberate parental actions ensured the maximum delay in noticing physical differences among girls by forbidding direct naked interaction with other uncut girls; the mother frequently claimed, "Your bodies are clean, keep it secret, meant only for husbands" (KI9). Despite this, seven participants eventually noticed bodily abnormalities; for some, this realization was delayed until adolescence (age 10 or older), prompted by sexual activity or peer exposure, as "at home, all the girls looked the same" (KI6)

Strategies to hide a sibling's cutting included sending other children away, then returning after it is done to create a fresh environment, with one witness recalling, "They removed us. When we returned, they barred us from seeing her naked. All care was strictly in the room; only adults could care for her. Mother said she was sick, and it was the doctor's order to prevent contact until she recovered. After three months, we were allowed to discover the healed scar" (KI4). In an unfavorable home environment, a pretence of medicalization was used to take the child away for cutting, claiming "she is sick and going to the traditional healer" (KI5).

**Theme 2: Blood and lies.** For those cut at an age of awareness, the experience was an 'inhumane process' of 'violence disguised as care,' characterized by force, restraint, and profound pain. The central deception framed this violence as a rite to become 'complete women.' Yet these traumatic memories fueled a powerful desire to break the intergenerational cycle, despite powerlessness. A KI recounted her forceful cutting experience:

**Table 4. Changes in FGM awareness from baseline to endline (Complete Sample, N=452).**

| Variable | | Baseline (n, %) | Endline (n, %) | p-value |
|---|---|---|---|---|
| Mean awareness score (SD) | | 44.08 (15.85) | 59.12 (16.85) | < 0.001† |
| FGM is harmful | | 412 (91.2) | 437 (96.7) | 0.001* |
| Harmful effects of FGM (Multiple responses allowed) | Excessive bleeding | 333 (80.8) | 400 (91.5) | |
| | Severe pain | 246 (59.7) | 357 (81.7) | |
| | Infections | 162 (39.3) | 311 (71.2) | |
| | Urinary problems | 124 (30.1) | 262 (60.0) | |
| | Psychological problems | 147 (35.7) | 231 (52.9) | |
| | Sexual problems | 137 (33.3) | 252 (57.7) | |
| | Difficult childbirth | 259 (62.9) | 257 (58.8) | |
| FGM is conducted under no medical reasoning | | 237 (52.4) | 419 (92.7) | |
| Sources of FGM awareness (Multiple responses allowed) | Learning from school | 281 (62.2) | 421 (93.1) | |
| | Mass media | 234 (51.8) | 132 (29.2) | |
| | Discussions with the community | 59 (13.1) | 73 (16.2) | |
| | Witnessing within the family, relatives, or neighbors | 30 (6.6) | 28 (6.2) | |
| | Positive FGM status (women only)^ | 17/234 (7.3) | — | |
| | Prior involvement in an anti-FGM program^ | 0 | — | |
| | Others (including hospital education, religious houses, and sexual relationships) | 3 | 7 | |
| FGM is still occurring in the community | | 179 (39.6) | 230 (50.9) | 0.002* |
| FGM violates the human rights of girls and women | | 387 (85.6) | 441 (97.6) | < 0.001* |
| Forms of violations associated with FGM (Multiple responses allowed) | Intimate partner violence | 185 (47.8) | 264 (58.4) | |
| | Child, early, and forced marriage | 176 (45.5) | 279 (61.7) | |
| | Stigma and gender discrimination | 155 (40.1) | 251 (55.3) | |
| | Right to health, security, and physical integrity | 108 (27.9) | 249 (55.1) | |
| | Right to life in cases of death | 164 (42.4) | 276 (61.1) | |
| | Right to be free from torture and cruel, inhuman, or degrading treatment | 97 (25.1) | 153 (33.8%) | |
| FGM is criminalized by national and international laws | | 394 (87.2) | 437 (96.7) | |
| Knew the 'International Day of Zero Tolerance for FGM, February 6', and its core aim | | 192 (42.5) | 424 (93.8) | |
| Desired FGM abandonment | | 367 (81.2) | 433 (95.8) | <.001* |

*McNemar's test; †Paired t-test; ^Assessed only at baseline to establish reporting patterns of FGM status or exposure.

When I was around six, my mother took me to my grandmother's village and found two aunties there as well. I didn't know their plan. The next morning, they woke me up to bathe. A while ago, an unknown older woman came; they boiled medicine and whispered inside for minutes, then I was called. Upon entry, the grandmother said, 'We are making you a clean and complete woman; you will be cut, do not worry.' I was scared and tried to run, but I was grabbed. They covered my face and mouth and held me down. The older woman spat the medicine all over my body and forcibly cut me. Nobody came to my aid. I fainted from the pain. When I woke up, they caressed me, saying, 'You are now a complete woman, the pain will be short, be strong.' My perineum was full of scars from the forceful cutting. I was in indescribable pain for over a month, especially during urination. The pain I went through was not of this world. I hate everyone every time I think about it... The frequent complaint of my husband's sexual dissatisfaction and bullying is

**Table 5. Observed FGM prevalence, types, and tribal distribution among delivering mothers in Chamwino District hospitals, June 2023 – February 2024.**

| Metric | Chamwino District Hospital | Dabalo Health Centre | Haneti Health Centre | Mpwayungu Health Centre | Chamwino Health Centre | Total |
|---|---|---|---|---|---|---|
| **Health Facility** | | | | | | |
| All mothers | | | | | | |
| Total mothers (N) | 1168 | 248 | 524 | 568 | 1262 | 3770 |
| Total FGM, n (%) | 132 (11.3) | 40 (16.1) | 270 (51.5) | 122 (21.5) | 60 (4.8) | 624 (16.6) |
| Young mothers (15–19 years) | | | | | | |
| Total Mothers (N) | 161 | 18 | 75 | 76 | 180 | 510 |
| FGM cases, n (%) | 42 (26.1) | 6 (33.3) | 56 (74.7) | 18 (23.7) | 36 (20.0) | 158 (31.0) |
| FGM typology (All positive mothers) | | | | | | |
| Type I, n | 12 | — | 4 | — | — | 16 |
| Type II, n (%) | 120 (90.9) | 40 (100) | 40 (100) | 122 (100) | 60 (100) | 608 (97.4) |
| Tribal distribution (All FGM cases) | | | | | | |
| Gogo, n (%) | | | | | | 459 (73.6) |
| Others, n (%) | | | | | | 165 (26.4) |

> also killing me. I feel like I came to this world to suffer. At times, I feel miserable and detest my body…the cup I should drink for the rest of my life…Although I lack power…I would not want my child to go through the same cup (moment of sorrow and tears) (KI2).

**Theme 3: Intergenerational revolt.** A central contradiction emerged between participants' awareness of FGM's harms and their profound sense of powerlessness to stop it. This "awareness-power gap" stems from a lack of authority, pressure to uphold norms, lack of uncertainty, economic dependence, and fear of social sanctions. As one explained, "I cannot stop the elders, if it's to follow traditions...I don't even remember what was done to me" (KI2), while another noted that young husbands, seen as "children with nothing," also lack the power to object (KI3).

Participants identified that effective abandonment requires moving beyond individual awareness to address these systemic barriers. Their recommendations form a multi-pronged strategy: (1) Prioritize early, youth-focused education, as "children's perspectives are more malleable" (KI7); (2) Transform family and community engagement by ensuring "education combines partner involvement and extends to age-based community conferences" to build a united front (KI10, 17); (3) Alternative rites by marrying later with financial independence to "gain strength" (KI3); and Enforce legal accountability and proactive surveillance. They argued for a "strategy focused on law enforcement and investigating all children" (KI3) and proposed "a tracking policy of all children from now on, with the option to sue offenders even after 20 years" to establish discouragement and accurately determine current prevalence (KI8).

Lastly, they highlighted the need to localize global efforts, with one noting she had "heard about World FGM Day but never knew what it exactly meant" (KI1), highlighting the gap between international campaigns and community-level awareness.

## Discussion

This study shows that a community-based, theory-informed intervention significantly improved FGM awareness among young adults in a high-prevalence Tanzanian district, confirmed by the increased mean awareness score (15.04 percentage points), with a (Cohen's $d = 0.65$). The intervention, implemented with high fidelity, achieved strong engagement and reach across schools, hospitals, and communities. A key indicator of sustainability is the champion network, organized

PLOS Global Public Health

**Table 6. Themes, categories, and illustrative codes.**

| Theme | Category | Illustrative Codes |
|---|---|---|
| The hidden system | Covert practices | Hidden rituals (e.g., structural concealment, silent preparation, and closed doors) |
| | | Community collusion (e.g., collective silence or cooperation to protect FGM practitioners) |
| | | Paradigm shift in early childhood |
| | | Ceremonial extinct |
| | | Medicalized secrecy |
| | Patriarchal Delegation | Father's passive role |
| | | Elder Female perpetuation |
| | Post-cutting con-cealment tactics | Deception cloaked in sweet lies |
| | | Taboo discussion |
| | | Enforced isolation |
| Blood and lies | | Inhumane process (ritualized violence) |
| | | Endless trauma bonding |
| | | Generational trauma cycle-breaking desire |
| Intergenera-tional revolt | Individual fear and silence | Economic coercion |
| | | Decision power deficit |
| | | Culture upholds |
| | System deficit | Structural gap in community-specific FGM abandonment programs |
| | | Legal deterrence limits |
| | | Education gaps |
| | | Lack of collective commitment at the com-munity level |
| | Sustainable efforts | Child Surveillance |
| | | Community-tailored anti-FGM program |
| | | Legal enforcement |
| | | Enhance FGM integration in perinatal care |
| | | Commitment from leaders |
| | | Changing agent role |
| | | Alternative rites of passage |

through a WhatsApp group and health clubs, which remained active beyond the study period and became a community-led platform for ongoing FGM abandonment activities and health education.

The significant quantitative changes confirm the intervention's success in improving individual-level mediators (awareness, perceived risk, and intention). However, finding integration reveals a complex reality: a resilient "hidden system" and a critical "awareness-power gap." The qualitative findings demonstrate that these individual-level changes are insufficient to overcome community-level structural barriers, indicating a breakdown from individual intention to collective abandonment. This suggests that awareness-raising alone is insufficient without strategies targeting structural power dynamics, a finding consistent with global evidence that community-level interventions often shift attitudes without achieving behaviour change [4].

**Comparison with national and regional data**

The 2022 TDHS-MIS reported a 47% FGM prevalence in Dodoma, much higher than the 16.6% in the Chamwino clinical audit [2], likely due to sample differences. The 16.6% reflects prevalence among mothers aged 13–41 across five

facilities, with young mothers (15–19 years) showing a 31.0% rate, nearly double the overall. Beyond sample composition, geographic heterogeneity likely contributes: Chamwino is a predominantly rural district with distinct socio-demographic characteristics, and our sample was younger (13–41 years) than the TDHS-MIS reproductive-age cohort (15–49 years).

This disproportionate burden on young mothers, warrants reflection. Several explanations are plausible. First, this may reflect a cohort effect: younger women may have been cut more recently, during a period when the practice shifted from public ceremonies to early childhood secrecy to evade laws [30]. Second, the finding aligns with global evidence that FGM prevalence varies by age cohort, with younger women sometimes showing higher rates where practices remain deeply hidden [31]. Third, the gap between self-reported (7.3%) and observed (31.0%) prevalence highlights underreporting, driven by fear of legal consequences and social desirability bias [31,32].

Notably, self-reported FGM (7.3%) closely matched the national average of 8% [2], highlighting consistent under-reporting in surveys [31]. This substantial discrepancy between self-reported (7.3%) and clinically observed (16.6%) prevalence, and even more so for young mothers (31.0%), has significant implications for surveillance. Self-reporting often underestimates true prevalence, particularly in settings where the practice is illegal [31], risking ineffective interventions. Our study used clinical audits as a complementary approach to capture FGM prevalence in high-burden settings.

Evidence from prevalent countries shows that 7 out of 10 individuals wish for FGM abandonment [33], aligning with our 95.8% endline findings. However, qualitative data revealed that this near-universal desire coexists with an "awareness-power gap", people may support abandonment yet feel powerless to enact it, a tone not captured in national surveys [4].

### Triangulating evidence: Awareness gains amidst a hidden system

The quantitative results are promising: the mean awareness score increased by 34% relative to baseline, and near-universal desire for abandonment at endline, aligning with successful community-engaged strategies [4,23,34]. The integrated theoretical framework likely boosted awareness, self-efficacy, and social norms. Yet a contradiction persists: despite these awareness gains, the qualitative theme "The Hidden System" reveals that FGM practices have shifted to early childhood and deepened into secrecy to evade laws [30]. As participants described, "now babies are cut discreetly very young" (KI6), and "practitioners disguise themselves as traditional healers... their identities are kept classified among a few elders" (KI10). This covert adaptation is reflected by 60.9% of those aware of FGM continuation, who could not explain why it persists, a powerful silencing social norm [35]. This highlights the need for targeted interventions such as this study, consistent with WHO (2025) guidance, emphasizing educational interventions for FGM-affected communities [6].

### The awareness-power gap: When awareness lacks agency

The critical synthesis emerges from the tension between improved attitudes and the 'Intergenerational Revolt' theme. While 95.8% desired abandonment, they described profound powerlessness due to economic dependence, fear of ostracization, and patriarchal control. This "awareness-power gap" indicates that the primary barrier is not intention but a lack of executable agency [36]. Findings revealed awareness of harms was weakly linked to acknowledgement of FGM continuation ($\varphi = 0.114$) and not significantly tied to desire for abandonment ($p = 0.073$). This highlights the gap between factual awareness and perceived social norms [26] and aligns with findings that community interventions may shift attitudes but not behaviors, necessitating ongoing efforts [4]. Men (57.9%) were more likely to acknowledge FGM continuation than women (43.8%). Our qualitative data and [32] explained this through patriarchal delegation: fathers playing a passive role, viewing it as "a woman's issue," which allows them to acknowledge it more openly due to perceived low risk, while women face social pressure to conceal it. Therefore, successful abandonment strategies should incorporate men as central figures [32,37,38], as employed in this study.

## Implications for policy and practice: A multi-level abandonment strategy

Our integrated findings argue for an evolution in abandonment strategies that moves beyond isolated awareness campaigns, aligning with WHO (2025) recommendations for comprehensive health sector engagement [6]:

1. Bridging the gap with programs combining education with activities fostering youth agency, like economic skills, legal literacy, abstaining from child marriage, and leadership. Engaging men and boys as accountable stakeholders, not just allies, is vital to dismantling patriarchal norms that uphold the practice [32,38].

2. Adapting interventions to covert reality with new mechanisms, including training health workers to identify and counsel at-risk families and to create safe, confidential reporting channels that protect whistleblowers, a capacity building recommended by the WHO guideline [6].

3. Implementing a synchronized ecological approach: sustainable abandonment requires coordinated actions at all levels, child surveillance and punishment (national), legal enforcement and local accountability (societal/community), intergenerational dialogue (interpersonal), and integrating FGM care and education into schools and health services (individual) [6,17].

4. Costs and scalability: The intervention used existing community structures (schools, health facilities, households, and local leaders), reducing the need for new infrastructure. Champions were volunteers from within the community, and activities were integrated into routine health and education settings. Instead of relying on external funding, this model aligns with existing domestically funded services, indicating potential for scaling within school-based health programs, reproductive health services, and community health worker networks. A formal cost-effectiveness analysis is needed to determine the resources required and identify the most cost-efficient components for sustained integration into government systems.

## Strengths, limitations, and generalizability

Key strengths include the mixed-methods design, community-based participatory approach, theory-informed intervention, and intervention-naïve sample. The main limitation is the single-arm, baseline-endline design, which cannot definitively establish causality; observed changes may be influenced by secular trends, social desirability bias, or community-wide diffusion of educational messages. Because this was a community-wide campaign with multiple exposure points, we did not track individual intervention attendance, precluding a dose-response analysis. Future efficacy studies could implement controlled exposure tracking. The qualitative findings, though in-depth, are solely from FGM-positive young mothers and lack perspectives of other key groups (e.g., uncut women, elders, practitioners, men).

The 19-point awareness scale was feasible to administer, with an average completion time of 10–12 minutes and a 96.6% retention rate, indicating acceptability among participants. The scale demonstrated acceptable internal consistency (Cronbach's α = 0.70) and was grounded in established literature and global guidance. However, based on our findings and systematic reviews [4], future adaptations could consider: [1] shortening the scale to reduce respondent burden while preserving core items; [2] adding items that probe reasons for FGM persistence more directly; and [3] integrating qualitative sub-questions to contextualize quantitative responses, as done in this study.

Our findings' generalizability is contextual. The sample, selected through a rigorous multi-stage strategy in purposively chosen wards, reflects key demographics of this high-prevalence district. The intervention model, which uses public institutions, is suitable for similar Tanzanian settings and is supported by a high retention rate. No direct incentives were provided for participation, supporting the authenticity of responses and suggesting the model could be replicated without financial inducements. The follow-up period was 10 months, sufficient to measure changes in awareness and attitudes but not long-term behavioral outcomes or FGM prevalence reduction. However, because wards were purposively selected,

caution is warranted when applying exact effect sizes more broadly. Nonetheless, the core finding, that awareness gains confront resilient structural barriers, likely applies to other contexts where FGM is a hidden norm, subject to tailored strategies.

## Conclusion

This study showed that a young-focused, theory-based program can change awareness and shift attitudes where none previously existed. It also uncovered the practice's hidden adaptive nature and how structural powerlessness can weaken positive attitudes. To abandon FGM, a dual approach is needed: ongoing education and empowerment of youth, along with dismantling covert structures through legal, economic, and community actions.

## Supporting information

**S1 Checklist. TREND Statement Checklist.**
(PDF)

**S2 Checklist. COREQ Checklist.**
(PDF)

**S1 Text. Training manual.**
(PDF)

**S1 Data. Data collection instruments.**
(PDF)

**S1 Table. Detailed participant tribes involved in the study baseline (N = 468) and endline (N = 452), Chamwino District, Tanzania.**
(PDF)

**S2 Table. Baseline and endline awareness findings stratified by sex characteristics (complete sample, N = 452).**
(PDF)

## Acknowledgments

We thank all members of the Chamwino community for their personal support and involvement in the study.

## Author contributions

**Conceptualization:** Leah Barthalome Kimario, Manji Nyaganya Isack, Beatrice Zakaria Temba, Agnes Cyril Msoka, Blandina Theophil Mmbaga.

**Data curation:** Leah Barthalome Kimario, Manji Nyaganya Isack.

**Formal analysis:** Leah Barthalome Kimario, Manji Nyaganya Isack.

**Funding acquisition:** Blandina Theophil Mmbaga.

**Investigation:** Leah Barthalome Kimario, Manji Nyaganya Isack.

**Methodology:** Leah Barthalome Kimario, Manji Nyaganya Isack, Beatrice Zakaria Temba, Agnes Cyril Msoka, Blandina Theophil Mmbaga.

**Project administration:** Blandina Theophil Mmbaga.

**Resources:** Leah Barthalome Kimario.

**Supervision:** Blandina Theophil Mmbaga.

**Validation:** Leah Barthalome Kimario, Manji Nyaganya Isack, Beatrice Zakaria Temba, Agnes Cyril Msoka, Blandina Theophil Mmbaga.

**Writing – original draft:** Leah Barthalome Kimario, Manji Nyaganya Isack.

**Writing – review & editing:** Beatrice Zakaria Temba, Agnes Cyril Msoka, Blandina Theophil Mmbaga.

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
