## [Decision Letter · Decision Letter 0]

26 Nov 2025

PGPH-D-25-02903

Targeting a future generation free from female genital mutilation: an awareness impacting observational study in central Tanzania

Dear Dr. Kimario,

Thank you for submitting your manuscript to PLOS Global Public Health. After careful consideration, we feel that it has merit but does not fully meet PLOS Global Public Health’s publication criteria as it currently stands. Therefore, we invite you to submit a revised version of the manuscript that addresses the points raised during the review process.

Please note that we have only been able to secure a single reviewer to assess your manuscript. We are issuing a decision on your manuscript at this point to prevent further delays in the evaluation of your manuscript. Please be aware that the editor who handles your revised manuscript might find it necessary to invite additional reviewers to assess this work once the revised manuscript is submitted. However, we will aim to proceed on the basis of this single review if possible.

The reviewer's comments are available below and they have provided constructive feedback with requests for clarity on the study design, consistent reporting of results and a synthesis of the results in the discussion section among other comments. Please review their assessment and make the appropriate revisions to address the concerns raised.

We look forward to receiving your revised manuscript.

Kind regards,

Emma Campbell, Ph.D

Staff Editor

Journal Requirements:

Additional Editor Comments (if provided):

Reviewers' comments:

Reviewer's Responses to Questions

**Comments to the Author**

1. Does this manuscript meet PLOS Global Public Health’s publication criteria? Is the manuscript technically sound, and do the data support the conclusions? The manuscript must describe methodologically and ethically rigorous research with conclusions that are appropriately drawn based on the data presented.

Reviewer #1: Yes

2. Has the statistical analysis been performed appropriately and rigorously?

Reviewer #1: Yes

3. Have the authors made all data underlying the findings in their manuscript fully available (please refer to the Data Availability Statement at the start of the manuscript PDF file)?

Reviewer #1: Yes

4. Is the manuscript presented in an intelligible fashion and written in standard English?

Reviewer #1: Yes

5. Review Comments to the Author

Reviewer #1: The manuscript addresses an important and highly sensitive public-health issue—female genital mutilation (FGM) in Tanzania—and evaluates a multi-component awareness intervention among adolescents and young women. The topic is relevant, ethically grounded, and socially impactful. The mixed-methods design strengthens the manuscript, and the qualitative narratives are powerful.

However, substantial revisions are required to improve clarity, methodological rigor, and scientific positioning.

The manuscript calls itself "mixed-methods cross-sectional," yet includes pre-post data and longitudinal follow-up. This is not cross-sectional.

Discuss and cite https://pubmed.ncbi.nlm.nih.gov/37539674/

Clarify the design: this appears to be quasi-experimental pre-post intervention with explanatory qualitative component.

The sampling approach is confusing; simplify and provide a CONSORT-style flow chart.

Some tests are not justified (e.g., paired t-test for awareness scores—were these normally distributed?).

Provide rationale for using prevalence ratios rather than risk ratios.

Many p-values reported without effect sizes in text.

Action:

State test assumptions and verify data suitability

Report effect sizes and confidence intervals consistently

Consider simplifying analytic presentation for readability

No true control group; intervention effect cannot be isolated from secular trends or social desirability bias.

Significant risk of contamination (e.g., community discussion and school-based diffusion).

Manuscript is lengthy and repetitive; consider streamlining:

Introduction too long—reduce general material, increase Tanzania-specific gaps

Methods repetitive in intervention description

Discussion reiterates results instead of synthesizing

6. PLOS authors have the option to publish the peer review history of their article (what does this mean?). If published, this will include your full peer review and any attached files.

**Do you want your identity to be public for this peer review?** For information about this choice, including consent withdrawal, please see our Privacy Policy.

Reviewer #1: No

Figure Resubmissions:

---

## [Decision Letter · Decision Letter 1]

23 Feb 2026

PGPH-D-25-02903R1

Targeting a future generation free from female genital mutilation: a mixed-methods quasi-experimental study of an awareness intervention in central Tanzania

Dear Dr. Kimario,

Thank you for submitting your manuscript to PLOS Global Public Health. After careful consideration, we feel that it has merit but does not fully meet PLOS Global Public Health’s publication criteria as it currently stands. Therefore, we invite you to submit a revised version of the manuscript that addresses the points raised during the review process.

The manuscript has been evaluated by two reviewers, and their comments are available below.

In addition to the concerns noted by reviewer 2 below, there are several other issues that must be addressed, mostly concerning the quantitative data collection tool and the analyses of these data.

1) You describe a 19-item knowledge scale, but the data collection instrument detailed in the S3 file contains 12 questions on Knowledge, Attitude, and Perceptions, some of which are noted as excluded from the awareness level estimation. It is not clear what the 19 items in the knowledge scale are. Please clarify what the 19 items are, and how each was scored. Please also consider consistency in terminology: is this measure Knowledge or Awareness?

2) Please provide baseline results as well as the endline results reported in Table 4.

3) The primary outcome is described as "the change in the proportion of participants with ‘Adequate Awareness’ of FGM health risks, defined as a score >76% on a 19-item knowledge scale, with scores categorized as inadequate (<50%), moderate (50–75%), and adequate (>76%)"

It is not at all clear what the basis for these categories is, or why the scores even should be categorised. You cite Kafle et al. (2020), but they provide no rationale or justification for the categorisation of continuous data. Unless there is a statistical/distributional reason (e.g., bimodal distribution of data), or a well-defined a priori cutoff (e.g., a physiological parameter that falls above/below a critical threshold), continuous data should not be divided into arbitrary categories.

Therefore, please could you treat your knowledge/awareness data as continuous in all analyses, which means replacing the modified Poisson regression and logistic regression with linear regression analyses.

Could you please carefully revise the manuscript to address all comments raised?

We look forward to receiving your revised manuscript.

Kind regards,

Steve Zimmerman, PhD

PLOS Staff Editor

Journal Requirements:

Additional Editor Comments (if provided):

Reviewers' comments:

Reviewer's Responses to Questions

**Comments to the Author**

1. If the authors have adequately addressed your comments raised in a previous round of review and you feel that this manuscript is now acceptable for publication, you may indicate that here to bypass the “Comments to the Author” section, enter your conflict of interest statement in the “Confidential to Editor” section, and submit your "Accept" recommendation.

Reviewer #1: All comments have been addressed

Reviewer #2: All comments have been addressed

2. Does this manuscript meet PLOS Global Public Health’s publication criteria? Is the manuscript technically sound, and do the data support the conclusions? The manuscript must describe methodologically and ethically rigorous research with conclusions that are appropriately drawn based on the data presented.

Reviewer #1: Yes

Reviewer #2: Yes

3. Has the statistical analysis been performed appropriately and rigorously?

Reviewer #1: Yes

Reviewer #2: I don't know

4. Have the authors made all data underlying the findings in their manuscript fully available (please refer to the Data Availability Statement at the start of the manuscript PDF file)?

Reviewer #1: Yes

Reviewer #2: No

5. Is the manuscript presented in an intelligible fashion and written in standard English?

Reviewer #1: Yes

Reviewer #2: Yes

6. Review Comments to the Author

Reviewer #1: Revised version: The authors addressed all the reviewes comments

Reviewer #2: This study aimed to examine knowledge change of an multipronged intervention as well as contextual analysis. It is well written and clear

A few comments:

1) The evidence on what works - systematic review of interventions of seminal publications references below) such are NOT referenced or referred to in the intervention design rationale or discussion.

a) Matanda DJ, Van Eekert N, Croce-Galis M, Gay J, Middelburg MJ HK. What interventions are effective to prevent or respond to female genital mutilation? A review of existing evidence from 2008–2020. PLOS Glob Public Heal

b) World Health Organization. (2025). WHO guideline on the prevention of female genital mutilation and clinical management of complications (ISBN 978-92-4-010728-1). World Health Organization. https://iris.who.int/handle/10665/381102

2) The intervention duration is not clear. How long was the exposure?

3)The authors should explain why 2022 TDHS-MIS data on FGM prevalence/ knowledge in Dodoma was not considered for baseline and also compare the survey findings on FGM prevalence, knowledge and attitudes with this study results.

4) The authors would need to explain why they chose "adequacy of knowledge of health risks" as success based on existing literature which does not support the rationale that "knowledge of health risks" is associated with behavioral/attitudinal or behavior change specifically on FGM

5) What was the definition of strong fidelity of implementation? Can this be elaborated?

6) The higher prevalence of FGM among young mothers (15 - 19) - almost double of overall FGM prevalence warrants an explanation/reflections

7) From a practicality point of view - is a 19 scale knowledge scale feasible to be conducted? Are the authors reconsidering or providing a recommendation given the findings and other systematic reviews whether this tool was adequate or requires revision?

8) Implications for policy or strategy, one key area is costs and scalability of this intervention. If the intervention is dependent on siloed approach dependent on external funds, chances that it is not scalable/affordable. Authors would need to speak to how this intervention could be part of existent domestically funded services that have a wide reach.

7. PLOS authors have the option to publish the peer review history of their article (what does this mean?). If published, this will include your full peer review and any attached files.

**Do you want your identity to be public for this peer review?** For information about this choice, including consent withdrawal, please see our Privacy Policy.

Reviewer #1: No

Reviewer #2: **Yes:** Wisal Ahmed

Figure Resubmissions:

---

## [Decision Letter · Decision Letter 2]

10 Apr 2026

Targeting a future generation free from female genital mutilation: a mixed-methods quasi-experimental study of an awareness intervention in central Tanzania

PGPH-D-25-02903R2

Dear Ms Kimario,

We are pleased to inform you that your manuscript 'Targeting a future generation free from female genital mutilation: a mixed-methods quasi-experimental study of an awareness intervention in central Tanzania' has been provisionally accepted for publication in PLOS Global Public Health.

Best regards,

Tanmay Bagade, Ph.D., MS (O&G), MPH, MHM

Academic Editor

Reviewer Comments (if any, and for reference):

Reviewer's Responses to Questions

**Comments to the Author**

1. If the authors have adequately addressed your comments raised in a previous round of review and you feel that this manuscript is now acceptable for publication, you may indicate that here to bypass the “Comments to the Author” section, enter your conflict of interest statement in the “Confidential to Editor” section, and submit your "Accept" recommendation.

Reviewer #1: All comments have been addressed

2. Does this manuscript meet PLOS Global Public Health’s publication criteria? Is the manuscript technically sound, and do the data support the conclusions? The manuscript must describe methodologically and ethically rigorous research with conclusions that are appropriately drawn based on the data presented.

Reviewer #1: Yes

3. Has the statistical analysis been performed appropriately and rigorously?

Reviewer #1: Yes

4. Have the authors made all data underlying the findings in their manuscript fully available (please refer to the Data Availability Statement at the start of the manuscript PDF file)?

Reviewer #1: Yes

5. Is the manuscript presented in an intelligible fashion and written in standard English?

Reviewer #1: Yes

6. Review Comments to the Author

Reviewer #1: revised version: the authors addressed all the reviewers comments

7. PLOS authors have the option to publish the peer review history of their article (what does this mean?). If published, this will include your full peer review and any attached files.

**Do you want your identity to be public for this peer review?** For information about this choice, including consent withdrawal, please see our Privacy Policy.

Reviewer #1: No
